# A High-Copy Suppressor Screen Reveals a Broad Role of Prefoldin-like Bud27 in the TOR Signaling Pathway in *Saccharomyces cerevisiae*

**DOI:** 10.3390/genes13050748

**Published:** 2022-04-24

**Authors:** Francisco Gutiérrez-Santiago, María Cintas-Galán, Manuel Martín-Expósito, Maria del Carmen Mota-Trujillo, Cristina Cobo-Huesa, Jorge Perez-Fernandez, Francisco Navarro Gómez

**Affiliations:** 1Departamento de Biología Experimental-Genética, Universidad de Jaén, Paraje de las Lagunillas, s/n, E-23071 Jaén, Spain; fgutierr@ujaen.es (F.G.-S.); mcintas@ujaen.es (M.C.-G.); mexposit@ujaen.es (M.M.-E.); mmota@ujaen.es (M.d.C.M.-T.); cch00008@red.ujaen.es (C.C.-H.); jpfernan@ujaen.es (J.P.-F.); 2Centro de Estudios Avanzados en Aceite de Oliva y Olivar, Universidad de Jaén, Paraje de las Lagunillas, s/n, E-23071 Jaén, Spain

**Keywords:** prefoldin-like, Bud27, TOR signaling pathway, *Saccharomyces cerevisiae*

## Abstract

Bud27 is a prefoldin-like, a member of the family of ATP-independent molecular chaperones that associates with RNA polymerases I, II, and III in *Saccharomyces cerevisiae*. Bud27 and its human ortholog URI perform several functions in the cytoplasm and the nucleus. Both proteins participate in the TOR signaling cascade by coordinating nutrient availability with gene expression, and lack of Bud27 partially mimics TOR pathway inactivation. Bud27 regulates the transcription of the three RNA polymerases to mediate the synthesis of ribosomal components for ribosome biogenesis through the TOR cascade. This work presents a high-copy suppression screening of the temperature sensitivity of the *bud27*Δ mutant. It shows that Bud27 influences different TOR-dependent processes. Our data also suggest that Bud27 can impact some of these TOR-dependent processes: cell wall integrity and autophagy induction.

## 1. Introduction

The TOR (Target of Rapamycin) signaling pathway is one of the most important mechanisms to control and coordinate cell growth and nutrient availability, and is conserved in all eukaryotes [1,2,3,4,5]. TOR acts as a stress sensor by responding to different stimuli, such as rapamycin or nutrient starvation [6,7,8,9,10,11,12]. The TOR pathway is constitutively active under favorable growth conditions to maintain ribosome production and cell growth. However, it is inhibited under stress conditions and ribosome biogenesis ceases [4,5,13,14].

It can be considered that the final function of TOR, in combination with PKA signaling, is the repression of the stress response. Some of the major regulators of the stress response in *S. cerevisiae*, which are controlled by the TOR and PKA signaling pathways, are Rho1 GTPase, Rim15, as well as transcription factors Msn2/4 and Gis1, which regulate about 150 genes by their association with the stress response element (STRE) located in the promoters of these genes [8,10]. Interestingly, either rapamycin treatment of cells or nitrogen starvation mediates responses that involve Msn2 [10,15].

TOR signaling represses autophagy, and TOR signaling inactivation by situations that induce stress, such as rapamycin or nitrogen starvation, induces autophagy [10,16]. Autophagy is essential for cells to survive under stressful conditions and is one of the most common responses of eukaryotic cells to bypass nutritional limitations [16,17,18].

In *S. cerevisiae*, two different TOR complexes exist, TORC1 and TORC2, which share some components [19]. TORC1 regulates ribosome biogenesis, nutrient transport, autophagy, and cell cycle progression, and is sensitive to the drug rapamycin. TORC2, which is unaffected by rapamycin treatment, participates in the cell cycle-dependent polarization of the actin cytoskeleton, endocytosis, and cell wall integrity (CWI) [20,21,22].

Bud27 in *S. cerevisiae*, and its human ortholog URI, have been described as components of the TOR signaling pathway by coordinating nutrient availability with gene expression [23,24,25,26]. However, whether Bud27 acts upstream or downstream the TOR kinase is still unknown. Bud27/URI are members of the prefoldin family of ATP-independent molecular chaperones [24], interact with the Rpb5 subunit of eukaryotic RNA polymerases, and participate in nuclear and cytoplasmic functions, such as the biogenesis of RNA pols, transcription, and translation, among others [23,25,26,27,28,29,30,31]. Notably, Bud27 inactivation induces a transcriptional response that partially mimics the transcriptional response due to TOR inactivation by rapamycin [25]. Bud27 has been recently shown to mediate ribosome biogenesis by regulating the activity of the three RNA pols and the synthesis of ribosomal components, likely involving the activity of TORC1 kinase [25].

Yet, whether lack of Bud27 only impacts TOR activity to regulate ribosome biogenesis or other TOR-dependent processes as well is unclear. To unravel the participation of Bud27 in the mechanisms governed by the TOR signaling pathway, we performed a high-copy suppressor screen on the *bud27*Δ mutant strain to rescue the temperature-sensitive phenotype of this mutant. Our results show that Bud27 may modulate the different processes regulated by the TOR signaling cascade and impact CWI, and likely autophagy.

## 2. Materials and Methods

### 2.1. Yeast Strains, Genetic Manipulations, Media, and Genetic Analyses

Common yeast media, growth conditions, and genetic techniques were used as described elsewhere [32]. For nitrogen starvation, the SD (-N) medium was used and contained 0.17% yeast nitrogen base without amino acids or ammonium sulfate, and 2% glucose.

Calcofluor white (Fluorescent Brightener 28; Sigma-Aldrich, Darmstadt, Germany) was used at the indicated concentrations.

The employed yeast strain was BY4741 and its derivative isogenic with *bud27*Δ mutation. 

The *pRS315-GFP-ATG8* (*CEN; LEU2*) plasmid was employed for the autophagy analysis (see below) [33].

Screening of a WT yeast genomic multicopy library. *bud27*Δ cells were transformed with a 2 µm-based multicopy yeast genomic DNA library constructed in plasmid pFL44L [34]. Colonies were selected in SD medium with appropriate requirements at 37 °C. In an effort to determine if suppression was plasmid-linked, cells were cured of plasmids by being grown on medium containing 5-FOA and screened for growth at 37 °C. Finally, plasmids were rescued from transformants, amplified in *Escherichia coli* and transformed into *bud27*Δ cells. The plasmids that allowed growth at 37 °C and/or 36 °C upon retransformation were selected and sequenced.

### 2.2. Chitin Staining with Calcofluor White and Fluorescence Microscopy

Chitin staining was performed in cells grown exponentially (OD~0.6–0.7) in SD minimal medium with the appropriate requirements. Cells were collected by centrifugation and resuspended in calcofluor white (Fluorescent Brightener 28; Sigma-Aldrich, Darmstadt, Germany) at 0.1 mg/mL concentration and incubated for 10 min at room temperature. Then cells were washed three times in distilled water. Slides were covered with Vectashield mounting solution (Vector Laboratories, San Francisco, CA, USA). Fluorescence intensity was scored with a fluorescence microscope (Olympus BX51).

### 2.3. Autophagy Detection

Autophagy progression was monitored by the immunological detection of the Gfp accumulation processed from Gfp–Atg8, which is delivered to the vacuole to be degraded upon autophagy induction [35]. Gfp moiety is very resistant to proteolysis compared to Atg8. For these assays, the WT and *bud27*Δ cells were transformed with the *pRS315-GFP-ATG8* (*CEN; LEU2*) plasmid to allow the expression of Gfp-Atg8. Cells were grown at 30 °C in SD minimal medium to the mid-log phase (OD_600_~0.6), which corresponded to experiment time 0. Then, cells were washed three times, diluted, and shifted to SD (-nitrogen) during a time course (up to 2 h). Proteins were precipitated with TCA from 1 mL of culture cells and analyzed by western blot using an anti-Gfp antibody (GFP (D5.1), 2956; Cell Signaling). Anti-phosphoglycerate kinase, Pgk1 (22C5D8; Invitrogen, Waltham, MA, USA) was used to detect Pgk1 as the internal control.

The intensities of the immunoreactive bands on western blots were quantified by densitometry using the IMAGE STUDIO LITE software from the images acquired with an office scanner.

## 3. Results and Discussion

### 3.1. Genetic Screening for Multicopy Suppressors of the Temperature-Sensitive Growth Defect of the bud27Δ Mutant

In order to search for genes that overcome the temperature sensitivity phenotype of the *bud27*Δ mutant, we performed high-copy suppressor screening. Cells were transformed with a 2 μm-based (pFL44L) multicopy genomic DNA library [34] and selected for growth at 37 °C. Suppressors were then identified and characterized as described in the Materials and Methods. Of the 100,000 independent transformants, 71 were able to grow at 37 °C, but at different growth rates depending on the suppressor (weak growth was observed for some). Accordingly, suppressors were defined as those able to overcome the temperature sensitivity of the *bud27*Δ mutant at 36 °C. The plasmids responsible for the suppression of the *bud27*Δ mutant phenotype were extracted and the presence of the wild-type (WT) *BUD27* gene was analyzed by PCR. As expected, six independent plasmids contained the *BUD27* gene, either alone or in combination with the *FRS2* gene. A search was done by a PCR analysis for genes *RPB5* and *RPB6*, whose overexpression overcomes the temperature sensitivity of the *bud27*Δ mutant [29]. Nine plasmids containing the *RPB5* gene and twenty-six containing the *RPB6* gene were identified (Figure 1).

Finally, we sequenced the fragments contained in 19 plasmids responsible for the suppression of the *bud27*Δ temperature-sensitivity phenotype. BLAST searches against the *S. cerevisiae* genome revealed that the *RPC17* gene, which codes for the Rpc17 subunit of RNA pol III, was one of the independent suppressors (Figure 1). This finding falls in line with the role of Bud27 in not only regulating the activity of the three RNA pols in coordination with TOR signaling [25], but also in mediating their cytoplasmic assembly [29]. The gene *SUN4*, which codes for Sun4, was also identified as a high-copy suppressor. Interestingly, this protein physically interacts with the member of the glycogen synthase kinase 3 (GSK3) family, Mck1 (https://www.yeastgenome.org, accessed on 1 April 2022), which mediates the phosphorylation of RNA pol III subunit Rpc53, which is important for enzyme activity regulation [36].

Another plasmid bore an insert containing the *RPL40A* gene, which codes for the large ribosomal protein Rpl40a. Notably, three plasmids contained inserts that harbored genes *RPL8A*, *RPL34A*, or *RPL33B*, which also code for large ribosomal proteins, among other genes. Subcloning genes *RPL8A*, *RPL34A*, or *RPL33B* corroborated their function as suppressors of the *bud27*Δ mutant growth phenotype at 36 °C. These genes also overcame the sensitivity of the *bud27*Δ cells to rapamycin, a drug that inhibits the TOR signaling pathway [29] (Figure 1). These data might be related to the roles of Bud27 in ribosome biogenesis [25] and/or translation initiation [31]. In agreement with the role of Bud27 in ribosome biogenesis, *SNR69,* which codes for the C/D box small nucleolar RNA (snoRNA) Snr69 [37], was also identified as a multicopy suppressor. 

*SMY2* overexpression acted as a high-copy suppressor. It codes for a GYF domain protein involved in COPII vesicle formation, which suppresses ribosome biogenesis defects [38] and influences the translation initiation of some mRNAs [39]. Two other multicopy suppressors are genes *FRS2* and *MSD1,* which, respectively, code for the α subunit of cytoplasmic phenylalanyl-tRNA synthetase Frs2 [40] and for mitochondrial aspartyl-tRNA synthetase Msd1 [41]. These results could well coincide with Bud27 participating in translation initiation [31], mediating ribosome biogenesis [25], or with a more general role of Bud27 and its human ortholog URI in coordinating nutrient availability with gene expression via the TOR signaling cascade [26]. 

*PAB1* has been identified as a multicopy suppressor. This gene codes for poly(A) binding protein Pab1, which performs many cellular functions associated with the 3-poly(A)-tail of messenger RNAs, including transport, translation, and mRNA decay [42,43,44,45]. These data agree with the role of Bud27 in translation and suggest a putative connection with mRNA degradation, as observed by the general decrease in mRNA stability in the *bud27*Δ mutant (Cuevas-Bermúdez et al., in preparation). Interestingly, Smy2 (see above) has been associated with the Ccr4-NOT deadenylase complex, which participates in mRNA degradation [46].

Another identified multicopy suppressor was *FLC3,* which codes for Flc3, a flavin adenine dinucleotide transporter [47] localized in the endoplasmic reticulum (ER) [48]. This protein interacts physically with Sac7 to render a non-active CWI pathway in the ER [47]. In addition, the deletion of *FLC* genes results in poor cell wall assembly [47]. Interestingly, Sun4 (see above) contributes to the regulation of cell wall morphogenesis and septation [49]. Furthermore, Smy2 (see above) has been described to be involved in the ER-dependent secretory pathway [46] as a multicopy suppressor of the *ptc1* mutant by showing defects for the CWI pathway [50]. Notably, TORC1 signaling acts in parallel with the unfolded protein response (UPR) to regulate ER stress and to modulate the CWI pathway [51]. Taken together, these results suggest a relation between Bud27 and cell wall assembly. Thus Bud27 might participate in CWI as a co-chaperone [24] member of the TOR signaling pathway [25,26].

In addition, sporulation has been shown to be related to CWI via the CWI MAP Kinase cascade [52]. In line with this, we identified *SPO20* as a multicopy suppressor in our screen, a gene that codes for the meiosis-specific subunit of the t-SNARE complex that is required for prospore formation during sporulation [53]. The *SSP2* gene was harbored in a plasmid capable of suppressing the phenotype of *bud27*Δ. *SSP2* codes for a sporulation-specific protein, Ssp2, which is localized in the spore wall and is required for sporulation after meiosis II [54]. Accordingly, we speculate that *SSP2* could be a suppressor of the *bud27*Δ phenotype. 

Sporulation in *S. cerevisiae* is a highly regulated process that is divided into three phases, with meiosis I and II taking place in the early and middle phases [55]. In line with the identified suppressors related to sporulation, a plasmid containing a fragment with genes *RMR1*, *ZIP2*, and *RME3*, which are all involved in meiosis, suppressed the temperature sensitivity phenotype of the *bud27*Δ mutant. Rmr1 is a protein required for meiotic recombination and gene conversion [56], Zip2 is a meiosis-specific protein involved in synaptonemal complex formation [57], while *RME3* codes an antisense transcript that represses *ZIP2* gene expression [58].

*GCR1* appeared as one of the strongest multicopy suppressors and was able to overcome the temperature sensitivity of the *bud27*Δ mutant, but also the sensitivity to drugs that affects translation (cycloheximide) and transcription elongation in *S. cerevisiae* (6-Azauracil and micophenolic acid [25,59]) (not shown). Gcr1 codes for a transcription factor that regulates the expression of glycolysis and ribosomal protein genes, RNA pol II transcription and the cell cycle [60,61]. Gcr1 has been reported to act in conjunction with transcription factor Rap1, which regulates the TOR signaling pathway [61,62] and could be involved in rRNA synthesis via Hmo1/Rap1 [63]. In agreement, *GCR1* overexpression overcame the sensitivity of the *bud27*Δ cells to rapamycin (Figure 1). Interestingly, the *gcr1*Δ mutant presents a defective vacuolar structure and affects autophagy in *S. cerevisiae* [61].

The *ATG39* gene has been identified as a multicopy suppressor. It codes for *Atg39*, a protein localized in the perinuclear ER that acts as a receptor for the selective autophagy of the ER and the nucleus in *S. cerevisiae*, a process known as nucleophagy [64]. Autophagy is associated with the TOR signaling cascade [10,16,17,18] and is related to ER stress [65], which compromises CWI [51]. Furthermore, the transcription of some autophagy-related genes has been shown to be altered after Congo Red or Zymolyase treatment in *S. cerevisiae*, two agents that induce cell wall damage [66]. Notably, a crosstalk between autophagy and sporulation has been described, two processes that correlate with meiosis [67]. Based on the above results, and on Bud27 participating in TOR signaling by coordinating nutrient availability with gene expression [25,26], we speculate that autophagy could be altered by lack of Bud27. Thus, it is tempting to speculate that genes *ATG36* and *PEX8*, which are involved in pexophagy [68,69,70,71] and are contained in two of the identified plasmids, could also act as suppressors of the *bud27*Δ growth phenotype. In line with the notion that autophagy was altered in *bud27*Δ cells, the Mlt1 cell wall receptor, which belongs to the CWI pathway and allows autophagy activation as a response to nutrient deprivation [72], was down-regulated in the *bud27*Δ mutant [25]. 

### 3.2. Transcriptomic Analyses Reinforce the Role of Bud27 in the TOR Signaling Pathway and in Cell Wall Integrity

The cells lacking Bud27 trigger a transcriptional response that partially mimics the repression of the TOR signaling pathway [25] and resembles previous data from rapamycin-treated cells [7,19,73]. In agreement, some of the identified suppressors overcame rapamycin sensitivity of the *bud27*Δ mutant (Figure 1).

By using the previously described data [25], we identified 719 genes that were differentially affected by Bud27 inactivation in relation to rapamycin treatment (288 up-regulated and 431 down-regulated). The STRING analysis [74] of the GO categories for the differentially up-regulated genes revealed biological processes that corresponded mainly to biosynthetic process (Appendix A). Contrarily and interestingly, the GO categories for the differentially down-regulated genes were mainly related to cell wall organization and cell wall biogenesis (Appendix A). 

Taken together, these results reinforce the role of Bud27 in the TOR pathway and the possibility that lack of Bud27 may affect CWI.

These data led us to wonder whether lack of Bud27 could affect CWI. Then, we analyzed the growth of the *bud27*Δ mutant and its isogenic wild-type (WT) strain in YPD-rich medium and the SD minimal medium containing calcofluor white, a chitin-binding agent that can inhibit the growth of cells with an abnormally large amount of chitin [75]. As shown in Figure 2A for growth in SD minimal medium, *bud27*Δ cells were sensitive to calcofluor white. In contrast, the WT cells did not display such sensitivity (similar results were obtained in the YPD-rich medium). Then, we examined cells by fluorescence microscopy after calcofluor white staining. We found that the *bud27*Δ mutant cells showed brighter fluorescence (Figure 2B), which indicates excess chitin. This finding falls in line with data previously observed for other mutants with cell wall defects [75,76,77,78,79]. 

Altogether, these data indicate that lack of Bud27 leads to stress, which affects CWI. In line with this, some of the suppressors (*SPO20*, *SMY2,* the fragment containing *RMR1, ZIP2* and *RME3, SUN4*, and the putative suppressor *SSP2*) slightly overcame the sensitivity of the *bud27*Δ mutant cells to calcofluor (Appendix A).

### 3.3. Lack of Bud27 May Cause Slower Autophagy Induction

As the above results suggest a relation between Bud27 and autophagy, we investigated whether this process could be altered in the *bud27*Δ mutant cells. For this purpose, we transformed the *bud27*Δ mutant and its WT isogenic strain with a plasmid harboring a *GFP-ATG8* construction coding for Gfp-Atg8 [33]. Atg8 is a member of the ubiquitin-like family proteins, and has been identified as one of the core elements in autophagy that bind to autophagic receptors and recruit cargo proteins for degradation [33,80]. During autophagy induction, Gfp is processed from Gfp-Atg8 and fragmentation can be analyzed by western blot [33,72]. Our results showed higher levels of free (processed) Gfp with respect to the unprocessed form (Gfp-Atg8) at time 0, in the *bud27*Δ mutant cells. Moreover, processing of Gfp-Atg8 to free Gfp seemed to occur more slowly in the *bud27*Δ mutant cells with respect to the WT cells during the time course under nitrogen starvation, reaching lower levels of free Gfp vs. Gfp-Atg8 after 120 min (Figure 3). 

These results suggest that autophagy is active in the absence of Bud27, probably because of TOR signaling pathway inactivation [25]. In addition, autophagy induction may occur more slowly, likely by dysregulation of TOR pathway. The role of Bud27, likely downstream of TORC1 and TORC2, could impact TOR regulation affecting different processes, such as autophagy, CWI, transcription, or ribosome biogenesis, among others. The influence of Bud27 in TOR regulation could complement and not be contradictory with the negative effects that Tor1 hyperactivation has on autophagy induction and in cell wall [51].

## 4. Conclusions

Our work reinforces the role of Bud27 in the TOR signaling cascade by showing that it may broadly impact different TOR-dependent processes. The results herein reported indicate a new role for Bud27 in influencing CWI. Finally, our data suggest that lack of Bud27 affects autophagy induction, which probably results from TOR pathway alteration. Taken together, our data suggest a functional relation of Bud27 with TORC1 and TORC2, according to the main actions of these complexes in autophagy and CWI regulation [35,51,72].

## Figures and Tables

**Figure 1 genes-13-00748-f001:**
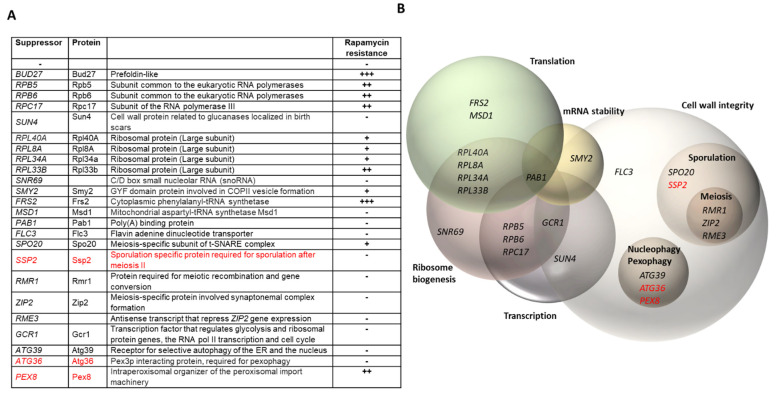
Suppressors of the temperature sensitivity phenotype of the *bud27*Δ mutant. (**A**) List of suppressors. Rapamycin sensitivity was strongly (+++), moderately (++), weakly (+), or not (−) overcome by suppressors. (**B**) Possible functional interactions between suppressors. Genes are grouped by their GO-Terms, and related cellular processes are enclosed together. Intersections indicate a functional or genetic relation. Red denotes genes that are putative suppressors.

**Figure 2 genes-13-00748-f002:**
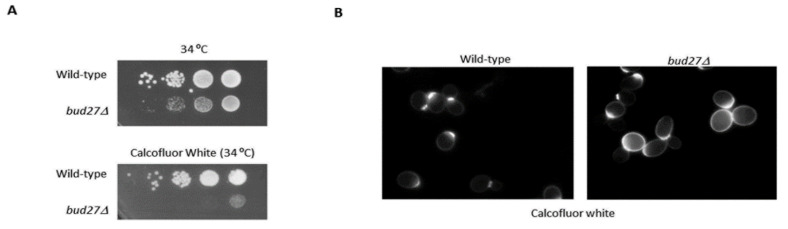
The *bud27*Δ mutant seems to affect cell wall integrity. (**A**) Growth of the wild-type and *bud27*Δ mutant at 34 °C in SD minimal medium w/o or with calcofluor white (10 μg/mL). (**B**) The yeast cells grown in minimal medium were stained with calcofluor white (0.1 mg/mL) and analyzed by fluorescence microscopy.

**Figure 3 genes-13-00748-f003:**
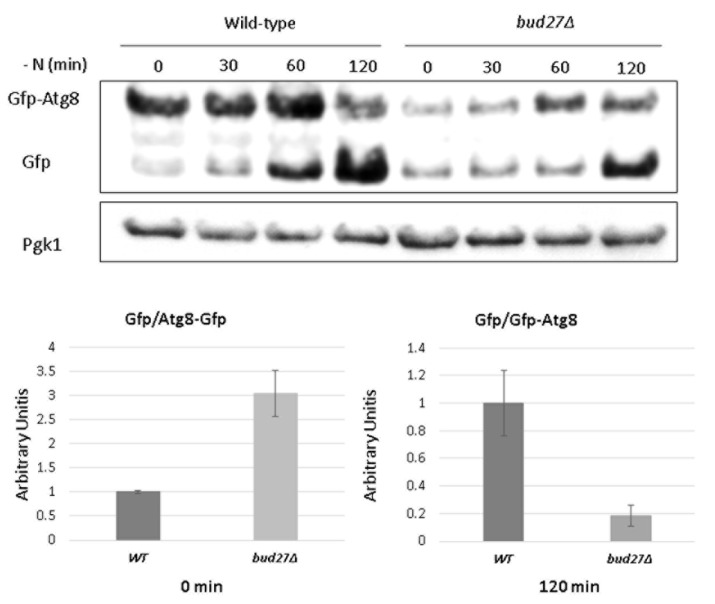
Gfp levels from Gfp-Atg8 degradation in the *bud27*Δ mutant seems to reflect lower autophagy induction. Upper panel: The wild-type and *bud27*Δ cultures transformed with the *pRS315-GFP-ATG8* plasmid were grown to the log phase (OD_600_~0.6) in SD minimal medium at 30 °C. Cells were then shifted to SD (–N) medium for the indicated time course. Aliquots were collected for protein extraction at the indicated times, and western blots with the anti-Gfp antibody was performed. The anti-Pgk1 antibody was used to detect Pgk1 as the internal control. Lower panel: Quantification of western blots signals shown in upper panel, corresponding to Gfp/Gfp-Atg8 obtained from wild-type and *bud27*Δ cultures. Graphs represent median and standard deviation of two independent biological replicates.

## Data Availability

Not applicable.

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
