# Peer review of "A High-Copy Suppressor Screen Reveals a Broad Role of Prefoldin-like Bud27 in the TOR Signaling Pathway in Saccharomyces cerevisiae"

_genes, 2022, doi:10.3390/genes13050748_

Round 1
Reviewer 1 Report
The authors performed a multicopy suppressor screen and identify a list of suppressors that overcome the temperature sensitivity of the bud27 mutant at 36°C. They discuss the possible involvement of Bud27 in various processes, based on the data. Autophagy is altered in the bud27 mutant. The authors use previously published data to identify genes that are differentially expressed when Bud27 is inactivated. The downregualted genes mainly relate to cell wall organization and biogenisis. The data imply a role for Bud27 in cell wall integrity. In support of this, the bud27 mutant was highly sensitive to calcofluor white. The bud27 mutant had reduced autophagy compared with the wild type. The authors suggest that bud27 affects autophay induction, possibly through TOR pathway alteration.
The paper is well written and the subject matter has wide relevance. There were some minor grammatical errors:
54, 138, 194, 204, 210, 213, 216/217 S. cerevisiae should be italicised
72 "The common" should be "Common"
73 "stravation" should be "starvation"
87 E. coli should be italicised
94 Delete "The"
121 "grow" should be "growth"
127 "overcome"should be "overcomes"OR "overcame"
143/4 "the member of the glycogen synthase kinase (GSK3)" should be "a member of the glycogen synthase kinase (GSK3) family" OR "the glycogen synthase kinase (GSK3)"
236 "the biosynthetic" should be "biosynthetic"
245 Delete "the"
289-291 Author contributions seem to be rather author fellowships
Author Response
Reviewer 1:
We thank the reviewer for his/her considerations, which will help to improve our manuscript. Accordingly, we have rewritten the manuscript and modified some figures. All changes have been highlighted in red in the manuscript text file.
1) The paper is well written and the subject matter has wide relevance. There were some minor grammatical errors:
We apologize for these inconveniences. However, most of the errors found are due to Editorial edition. We have corrected all errors and included reviewer’s suggestions.
Reviewer 2 Report
Gutiérrez-Santiago et al present work presents a high copy suppression screening of the bud27 mutant. The results reinforce previous studies suggesting that Bud27 affect processes in common with TOR. The data is potentially interesting, but the evidence provided for a broad-role in TOR signalling pathway is not very compelling.
Major issues:
- Rapamycin sensitivity should have been assessed for some of the strongest multicopy suppressors of bud27 identified. Resistance to rapamycin would reinforce the claim that this is Tor pathway related.
- Was Tor1 identified as a suppressor of bud27? Is not clear from the introduction (or if it is known) if bud27 is upstream or downstream tor kinase in the tor pathway.
- Concerning Figure 3, authors state that free GFP levels are lower in the bud27 mutant (line 269). Free GFP levels is used to evaluate the autophagic flux and for that the ratio between processed and unprocessed bands should be used as a readout (see doi: 10.3390/cells6030023). Since the unprocessed band is also lower is not clear if there is a defect in flux. Anyway, quantification of the western-blot should be shown (it is mentioned in M&M).
- There are two cases of data not shown reporting specific data (line 226 and 254). These should be included as supporting material (or removed).
- Authors state in the abstract that bud27 mutant partially mimics tor pathway inactivation. However authors report two bud27 phenotypes, slow autophagy induction and cell wall defects that have been described for hyperactive tor1, not inactive (https://doi.org/10.1002/1873-3468.13463). This should be discussed.
- In conclusions, line 281, the different tor-dependent processes impacted by bud27 identified in this work should be clearly indicated/summarized. Autophagy should be one of those, but not CWI, since it was differentially affected when comparing bud27 to rapamycin treated cells. The way it is written in the abstract (lines 20-21) it seems both CWI and autophagy are affected processes additional to TOR-dependent processes.
- Since CWI is more associated to TORC2 activity, does it means bud27 can be involved in both TORC1 and TORC2 functions? Please discuss.
Minor issues:
The number of replicates in each experiment should be indicated.
The text style needs some improving (e.g. lines 239-240). In the abstract, the introduction is disproportionally big in relation to the aims, results and conclusions of the present work.
I guess it was a formatting problem, but many symbols, such as the delta in bud27 mutant disappeared.
Author Response
Reviewer 2:
We thank the reviewer for his/her considerations, which will help to improve our manuscript. Accordingly, we have rewritten the manuscript and modified some figures. All changes have been highlighted in red in the manuscript text file.
Rapamycin sensitivity should have been assessed for some of the strongest multicopy suppressors of bud27 identified. Resistance to rapamycin would reinforce the claim that this is Tor pathway related.
According to reviewer’s suggestion, we incorporate rapamycin sensitivity for all multicopy suppressors. Results are included in the manuscript and in the modified Figure 1.
- Was Tor1 identified as a suppressor of bud27? Is not clear from the introduction (or if it is known) if bud27 is upstream or downstream tor kinase in the tor pathway.
The comment raised by the reviewer is very interesting. We have not identified TOR1 as a suppressor of bud27D phenotype. In addition, preliminary data from our lab suggest that Bud27 could be downstream TOR kinase in the TOR pathway, but these data are not still corroborated.
According to reviewer’s suggestion, we have included this information in the introduction.
- Concerning Figure 3, authors state that free GFP levels are lower in the bud27 mutant (line 269). Free GFP levels is used to evaluate the autophagic flux and for that the ratio between processed and unprocessed bands should be used as a readout (see doi: 10.3390/cells6030023). Since the unprocessed band is also lower is not clear if there is a defect in flux. Anyway, quantification of the western-blot should be shown (it is mentioned in M&M).
According to reviewer’s suggestion, we have included quantification of the western-blot in the modified Figure 3.
- There are two cases of data not shown reporting specific data (line 226 and 254). These should be included as supporting material (or removed).
As the reviewer proposes, we have included this information in the new version of the manuscript.
- Authors state in the abstract that bud27 mutant partially mimics tor pathway inactivation. However authors report two bud27 phenotypes, slow autophagy induction and cell wall defects that have been described for hyperactive tor1, not inactive (https://doi.org/10.1002/1873-3468.13463). This should be discussed.
We found this remark very interesting and, accordingly, we have discussed it.
- In conclusions, line 281, the different tor-dependent processes impacted by bud27 identified in this work should be clearly indicated/summarized. Autophagy should be one of those, but not CWI, since it was differentially affected when comparing bud27 to rapamycin treated cells. The way it is written in the abstract (lines 20-21) it seems both CWI and autophagy are affected processes additional to TOR-dependent processes.
We thank the reviewer for this observation. Accordingly, we have modified the corresponding information.
- Since CWI is more associated to TORC2 activity, does it means bud27 can be involved in both TORC1 and TORC2 functions? Please discuss.
This assumption has been discussed.
Minor issues:
- The number of replicates in each experiment should be indicated.
This Information has been included in the figure legends..
- The text style needs some improving (e.g. lines 239-240). In the abstract, the introduction is disproportionally big in relation to the aims, results and conclusions of the present work.
- I guess it was a formatting problem, but many symbols, such as the delta in bud27 mutant disappeared.
We apologize for these inconveniences. Most of the errors found are due to Editorial edition. We have corrected all errors and reviewed the style.
The manuscript has been proofread by a professional translator who is a native English speaker specialized in scientific writing
Round 2
Reviewer 2 Report
I have no further comments for authors. My decision was accept as it is.